# From DNA Copy Number Gains and Tumor Dependencies to Novel Therapeutic Targets for High-Risk Neuroblastoma

**DOI:** 10.3390/jpm11121286

**Published:** 2021-12-03

**Authors:** Bieke Decaesteker, Kaat Durinck, Nadine Van Roy, Bram De Wilde, Christophe Van Neste, Stéphane Van Haver, Stephen Roberts, Katleen De Preter, Vanessa Vermeirssen, Frank Speleman

**Affiliations:** 1Department for Biomolecular Medicine, Ghent University, Medical Research Building (MRB1), Corneel Heymanslaan 10, B-9000 Ghent, Belgium; Bieke.Decaesteker@ugent.be (B.D.); Kaat.durinck@ugent.be (K.D.); Nadine.vanroy@ugent.be (N.V.R.); Bram.DeWilde@UGent.be (B.D.W.); Christophe.vanneste@ugent.be (C.V.N.); Stephane.VanHaver@ugent.be (S.V.H.); Katleen.DePreter@ugent.be (K.D.P.); Vanessa.Vermeirssen@ugent.be (V.V.); 2Department of Internal Medicine and Pediatrics, Ghent University Hospital, Corneel Heymanslaan 10, B-9000 Ghent, Belgium; 3Department of Pediatrics, Memorial Sloan Kettering Cancer Center, New York, NY 10065, USA; robertss@mskcc.org; 4Department of Biomedical Molecular Biology, Ghent University, Technologiepark 71, B-9052 Zwijnaarde, Belgium

**Keywords:** neuroblastoma, DNA copy number gains, *MYCN*, dependency, drug targets

## Abstract

Neuroblastoma is a pediatric tumor arising from the sympatho-adrenal lineage and a worldwide leading cause of childhood cancer-related deaths. About half of high-risk patients die from the disease while survivors suffer from multiple therapy-related side-effects. While neuroblastomas present with a low mutational burden, focal and large segmental DNA copy number aberrations are highly recurrent and associated with poor survival. It can be assumed that the affected chromosomal regions contain critical genes implicated in neuroblastoma biology and behavior. More specifically, evidence has emerged that several of these genes are implicated in tumor dependencies thus potentially providing novel therapeutic entry points. In this review, we briefly review the current status of recurrent DNA copy number aberrations in neuroblastoma and provide an overview of the genes affected by these genomic variants for which a direct role in neuroblastoma has been established. Several of these genes are implicated in networks that positively regulate *MYCN* expression or stability as well as cell cycle control and apoptosis. Finally, we summarize alternative approaches to identify and prioritize candidate copy-number driven dependency genes for neuroblastoma offering novel therapeutic opportunities.

## 1. Introduction

Neuroblastoma (NB) is a pediatric tumor arising from the sympathetic neuronal lineage with a remarkably variable clinical presentation. The majority of children diagnosed with NB below the age of 18 months present with low or intermediate risk disease and have an excellent prognosis (INRGSS stages L1,L2 and MS with favourable genomic features) [1,2,3]. Older children with NB typically present with high-risk disease and require multi-modal intensive therapy. Despite this aggressive treatment, half of these patients will succumb to their disease. Those that do survive often have long-lasting negative health effects. A special type of NB occurs at a very young age, but with a peculiar metastatic pattern including skin, liver, and bone marrow metastases (Stage MS) but which regresses or differentiates with minimal therapeutic intervention [3].

After decades of research insights are growing into the underlying genetic features of the complex heterogeneous clinical behavior of this enigmatic tumor. Amplification of the *MYCN* oncogene was one of the first discovered genetic alterations in NB [4,5]. It was rapidly recognized as an important prognostic indicator and critical determinant for therapeutic patient stratification. Additionally, DNA content measurements revealed near triploid DNA content in stage L1/2 and MS cases [6]. Following these early pioneering observations, additional chromosomal analyses, loss-of-heterozygosity studies, comparative genomic hybridization, DNA arrays and most recently DNA sequencing have provided a comprehensive view on NB genomes. The mutational landscape has been shown to be relatively silent with *ALK* mutations being the only substantially mutated target in almost 10% of cases at diagnosis [7,8,9,10,11], while in relapsed cases increase of RAS/MAPK pathway mutations was noted (including *ALK* mutations) [12,13,14]. In older NB patients (adolescents and young adults with a more indolent or chronic form of NB), roughly 30% of cases present with ATRX deletions (indels) or loss-of function missense mutations and in-frame fusions in rare cases [15,16,17,18]. TERT rearrangements and enhancer hijacking is found in 25% of NB patients [17,18].

In contrast to the overall paucity of mutations, low and high-risk NB presents with highly recurrent whole chromosome or segmental imbalances and focal copy number aberrations (CNAs) respectively [19,20]. This remarkably consistent pattern of chromosomal imbalances has led to the suggestion that dosage effects of genes residing on the affected chromosomal regions contribute to tumor biology. This is obvious for the prototypical *MYCN* oncogene which is amplified in 30% of the high-risk NB cases and which acts as a strong dependency factor [21]. In addition to this frequent amplification event, additional genes have been identified (described in more detail in this review) that are involved in rare high-level amplifications or focal gains and which were subsequently shown to contribute to tumor behavior and dependency. Interestingly, many of these rare amplicons affect loci that map to the highly recurrent common large segmental gains such as 2p- and 17q gain suggesting that these rare focal gains and amplifications may serve as “smoking guns” to identify novel and potentially druggable NB genes.

In this review, we will first give a short historical overview on the detection of DNA copy number abnormalities in NB. Next, we summarize and discuss the currently identified genes with an established role in NB biology which are located on regions affected by gains and/or amplifications and address their role in NB tumor dependency potential for druggable interventions. While progress has been made in deciphering which copy number affected genes could potentially contribute to NB tumor behavior, several important questions remain unanswered. First, exactly how does a single extra copy of a large chromosomal region which encompasses dozens or more gene loci impact on the NB tumor phenotype? Second, are there any genes that enhance the effect of dosage genes through additive or epistatic effects? Third, do large DNA copy number imbalances act as drivers of early tumor formation, or do they predominantly contribute to tumor maintenance and dependency at a later stage, or both? At least for the first question, we have additional evidence that DNA copy number gains are unlikely to be innocent bystanders. Indeed, in *MYCN* driven mouse tumors we also observed large segmental chromosome gains for a region of mouse chromosome 11 which is syntenic to human chromosome 17 [22]. Also, recent whole genome sequencing data indicate that the common imbalances are early events preceding the occurrence of mutations [23]. However, this only provides partial insight into how these copy number gains affect tumor behavior; to answer these questions new experimental approaches are needed. The last section of this review proposes a number of non-exhaustive experimental and analytical strategies that are currently emerging and may allow us to answer these questions. Given the paucity of options for targeted therapy based on mutations only, a better understanding of the critically involved copy number driven genes and their putative epistatic interactions may provide important insights for testing novel therapeutic strategies.

## 2. Recurrent Focal and Large Segmental DNA Copy Number Alterations in NB

In addition to the existence of double minutes and homogeneously staining regions resulting from high level *MYCN* amplification, the first cytogenetic studies also detected recurrent large distal 1p deletions [24]. Chromosome 1p deletions are usually associated with *MYCN* amplification but also occur in *MYCN* non-amplified high-risk NB (Figure 1) [25]. Subsequent loss-of-heterozygosity studies defined a critical 1p36 segment commonly affected by 1p deletions [26,27] as well as the discovery of additional recurrent deletions affecting 3p, 4p and 11q [28]. Using comparative genomic hybridization (CGH) and DNA arrays these 11q deletions were shown to predominantly occur in high-risk *MYCN* non-amplified NBs [29,30] (Figure 1). These analyses also allowed a comprehensive mapping of gains across the NB genomes [19,31,32,33]. Whole chromosome imbalances were invariably present in low-risk L1/2 and MS NB while highly recurrent segmental gains mark stage M NBs. The most frequently occurring segmental gain implicated a large distal section of the long arm of chromosome 17 with breakpoints typically distal to 17q21 [34,35,36,37] (Figure 1). Other recurrent gains affecting larger chromosome segments involve 1q, 2p, 7q, 11q13.3 and 12q of which some loci are also affected by higher level focal amplification [23]. Of note, the typical deletions and gains of large chromosome arm segments often result from one single unbalanced chromosomal translocation [34].

Segmental chromosomal gains in general, and chromosome 17q gain in particular, were found to be associated with poor survival [19,35,39]. To identify genomic aberrations associated with poor outcome in high-risk NB, we applied logistic and Cox proportional hazard regression on CNAs from 556 high-risk NBs obtained at diagnosis [33,38]. Using this approach, we identified two types of CNAs that are associated with an extremely poor prognosis: (1) distal chromosome 6q losses detected in 5.9% of patients and associated with a 10-year overall survival probability of only 3.4% and (2) amplifications of regions not encompassing the *MYCN* locus detected in 18.1% of patients [(2p25.1 encompassing the ODC1 locus (12 samples), 2p23.2 including *ALK* (5), 2p25.1 including GREB1/NTSR2 (4), 6q16.3 including *LIN28B* (3), 12q15 including *MDM2*(2), 12q13.3/14.1 including *CDK4* (2), 11q13.2/13.3 including MYEOV and *CCND1* (2), and 5p15.33 including TERT (2). Several other amplicons, including 8q24.21 encompassing the MYC gene (1 sample)] and associated with a 10-year overall survival probability of only 5.8% [38].

DNA arrays with increased resolution and ultimately whole genome sequencing allowed reliable detection of smaller focal copy number alterations or indels and paved the way for the identification of several novel genes critically involved in NB. Whole genome sequencing unveiled chromothripsis as a mechanism driving complex multi-locus rearrangements causing combined amplification, deletion, gain and structural variants affecting particular loci [40]. Circle-sequencing further dissected the landscape of small and large extra chromosomal DNA in NB (respectively eccDNA and ecDNA) [41,42]. Large ecDNAs detectable as double minutes encompass entire genes and are associated with elevated expression levels of amplified genes. A recent experimental study on *DHFR* amplification under methotrexate exposure revealed that chromothripsis not only drives ecDNA formation but is also implicated in further ecDNA evolution [43]. In NB, chromosomal circle integration was shown to preferentially affect tumor suppressor genes and thus may further shape genomic evolution of these cancer cells [42].

## 3. DNA Copy Number Affected Genes Supporting *MYCN* Activity

In an early DNA copy number analyses, we identified several focal aberrations affecting genes that were connected to *MYCN* regulation or activity [44]. Taken together, a picture emerges where DNA copy number affected regions seem to be embedded within the tumor genome under the selective pressure to increasingly support *MYCN* activity in these cells. Here below we describe in more detail the key copy number driven genes with an established role in regulating *MYCN* activity (Figure 2 and Figure 3).

### 3.1. ALK

The anaplastic lymphoma kinase gene, typically present as fusion genes in anaplastic large cell lymphoma and a subset of lung cancers, was also found to be implicated in NB development. Activating *ALK* genetic alterations occur in 8–10% of all NB cases, and are mostly missense mutations affecting the ATP-binding pocket, with *ALK* amplifications found in a small subset of tumors. In addition, germline variants of *ALK* are found in familial NB cases supporting a bona fide oncogenic driver effect for activating *ALK* mutations in humans [7,8,9,10,11,45]. In *ALK* amplified cases, *MYCN* is typically co-amplified [45]. Both genes are located on the short arm of chromosome 2 which is often gained in high-risk NB and *MYCN*/*ALK* co-amplification typically results from a chromothripsis event involving the chromosome 2 short arm [40]. Remarkably, in mouse and zebrafish, *ALK* mutations are not sufficient as a single lesion to drive NB formation but *ALK* activating mutations do significantly accelerate *MYCN* driven NB formation [46,47,48]. Further support for functional interaction between mutant *ALK* and *MYCN* was provided by neuroendocrine prostate cancer research where activated *ALK* was shown to cooperate with *MYCN* via Wnt/β-catenin signaling [49].

**Figure 2 jpm-11-01286-f002:**
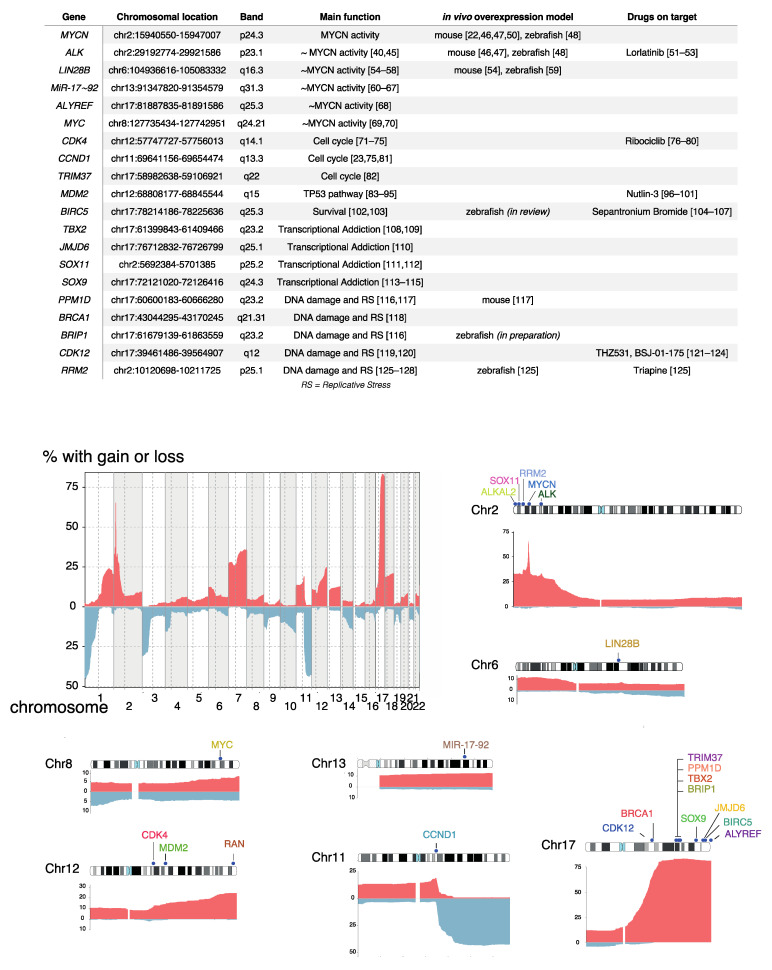
Table describing the copy number affected genes discussed in this review, their chromosomal location and cytoband, their main function with references, in vivo modeling with references, and drugs on target with references. Frequency of copy number gains/amplifications (red) and losses (blue) for chromosomes 1 to 22 in 542 high-risk NB samples with segmental CNAs [38]. Copy number affected genes in NB that were discussed in this review are depicted on their chromosome with the associated copy number frequency (chromosome 2, 6, 8, 11, 12, 13 and 17). Refs. [22,23,40,45,46,47,48,50,51,52,53,54,55,56,57,58,59,60,61,62,63,64,65,66,67,68,69,70,71,72,73,74,75,76,77,78,79,80,81,82,83,84,85,86,87,88,89,90,91,92,93,94,95,96,97,98,99,100,101,102,103,104,105,106,107,108,109,110,111,112,113,114,115,116,117,118,119,120,121,122,123,124,125,126,127,128] are cited in Figure 2.

Following knockdown or pharmacological inhibition of mutant *ALK* in NB, we performed transcriptome profiling and found that downstream signaling is mediated through the RAS/MAPK and PI3K/AKT pathways [129]. For each of these downstream signaling axes we identified a key component that could explain the increased aggressiveness of *MYCN*/*ALK* activated NB cells, i.e., (1) RAS/MAPK/ERK signaling activated the ETV5 transcription factor which controlled migration, invasion and colony formation in vitro and proliferation in a murine xenograft model ([130], p. 5) and (2) the PI3K/AKT/FOXO3a regulated ‘HMG-box transcription factor 1′ (HBP1) which inhibits both the transcriptional activating and repressing activity of *MYCN*. Of further interest, HBP1 itself is under negative control of *MYCN* through miR-17~92. Combined targeting of HBP1 by PI3K antagonists and *MYCN* signaling by BET- or HDAC-inhibitors blocks *MYCN* activity and significantly reduces tumor growth, suggesting a novel targeted therapy option for high-risk NB ([131], p. 1). Using a detailed proteomics analysis of the mutant *ALK* signaling components in NB, the signaling adaptor protein ‘insulin receptor substrate 2′ (IRS2) was also identified as a major *ALK* downstream target further controlling the PI3K/AKT/FOXO3a axis [132]. Finally, we also noted that mutant *ALK* upregulates RET and RET-driven sympathetic neuronal markers of the cholinergic lineage [129] which is intriguing in the light of the recent finding that malignant cells enriched in high-risk NB resemble a subtype of TRKB+ cholinergic progenitor population identified in the human post-natal gland [133].

At present, constitutive *ALK* activation through mutation or amplification is the most accessible target for precision medicine in NB [7,8,9,10,11] with a recently initiated clinical trial for the 3rd generation *ALK* inhibitor Lorlatinib [51,52,53]. Remarkably, the gene encoding the *ALK* ligand *ALKAL2* is located on the distal 2p segment (2p25.3). In view of the highly recurrent distal 2p gains in high-risk NBs this suggests that, in addition to increased expression levels of *MYCN* in cases of *MYCN* non-amplified tumors, an extra copy of the *ALK* gene and the gene for its ligand could impact tumor formation. The Palmer team recently investigated whether the *ALKAL2* ligand could potentiate NB progression in the absence of *ALK* mutation and showed that *ALKAL2* overexpression in mice drives *ALK* TKI-sensitive NB in the absence of *ALK* mutation. Consequently, both *ALK* mutated/amplified as well as 2p-gain high-risk NBs with elevated *ALK*/*ALKAL2* protein, may benefit from *ALK* tyrosine kinase inhibitor-based therapeutic intervention [134].

### 3.2. LIN28B

The discovery of *LIN28B* involvement in NB was triggered through the finding of a new amplified region on chromosome 6q21 in three high-risk NB tumors in a dataset of 263 tumors [54] as well as a GWAS study [54,55]. The smallest region of overlap (SRO) of the discovered amplicons in the first study encompassed four genes, including *LIN28B*. *LIN28B* overexpression was identified in three independent NB tumor series compared to normal reference tissues and a series of tissues from other malignancies. Most importantly, analysis of *LIN28B* expression showed pronounced overexpression in the tumors with *LIN28B* amplification while Kaplan-Meier analyses showed that high expression of *LIN28B* was significantly associated with poor overall survival in two cohorts of 88 and 283 patients with NB [54]. *LIN28B* is a key regulator of developmental processes by modulating microRNAs (miRNAs) of the let-7 family. *LIN28B* represses the let-7 miRNAs which consequently results in elevated *MYCN* protein expression in NB cells. Molenaar et al. further showed that *LIN28B*–let-7–*MYCN* signaling blocked differentiation of normal neuroblasts and NB cells. They recapitulated these findings in a mouse model in which *LIN28B* expression in the sympathetic adrenergic lineage induced development of NBs marked by low let-7 miRNA levels and high *MYCN* protein expression [54]. In a subsequent study, we performed a comprehensive analysis of the regulation of *LIN28B* in NB, with a specific focus on the contribution of miRNAs and showed that *MYCN* regulates *LIN28B* expression in NB tumors via two parallel mechanisms. First, using an unbiased *LIN28B*-3′UTR reporter screen, we found that miR-26a-5p and miR-26b-5p regulate *LIN28B* expression. We provided evidence that *MYCN* indirectly regulates the expression of miR-26a-5p, which on its turn reduces *LIN28B* levels, therefore establishing an *MYCN*-miR-26a-5p-*LIN28B* regulatory axis [56]. Second we demonstrated that *MYCN* regulates *LIN28B* expression via interaction with the *LIN28B* promoter, establishing a direct *MYCN*-*LIN28B* regulatory axis [56]. In addition, another study revealed ‘PDZ Binding Kinase’ (PBK) as a novel *LIN28B* target with implication in NB, repressed by let-7 and positively regulated by *MYCN* [57]. Further insights into the role of *LIN28B* in NB was provided by the Diskin team who identified the oncogene RAN, located on chromosome 12, as a *LIN28B* direct RNA target. They further showed that regional gains of chromosome 12q24 as an additional somatic alteration resulted in increased RAN expression. *LIN28B* influences RAN expression by promoting ‘RAN Binding Protein 2′ (RANBP2) expression in a let-7-dependent manner and by directly binding RAN mRNA. Aurora kinase A (AURKA) was found to be activated by both RAN and *LIN28B* through phosphorylation and let-7-mediated repression respectively [58]. Remarkably, the Look team discovered a let-7 independent function for *LIN28B* in NB development through the analysis of overexpression of wild-type (WT) *LIN28B* and a *LIN28B* mutant that is unable to inhibit let-7 processing. Unexpectedly, both increase the penetrance of *MYCN*-induced NB, potentiate the invasion and migration of transformed sympathetic neuroblasts, and drive distant metastases in vivo in zebrafish. Genome-wide chromatin immunoprecipitation coupled with DNA sequencing and co-immunoprecipitation experiments showed that *LIN28B* is recruited to active gene promoters in NB cells through protein-protein interaction with the sequence-specific zinc-finger transcription factor ZNF143 [59]. The activated downstream targets include transcription factors forming the adrenergic core regulatory circuitry that control the malignant cell state in NB [108,135,136] as well as GSK3B and L1CAM that are involved in neuronal cell adhesion and migration [59].

### 3.3. MiR-17~92

The oncogenic miR-17~92 cluster consists of six individual miRNAs (miR-17, miR-18a, miR-19a, miR-19b, miR-20a, and miR-92a) located within a polycistronic transcript on human chromosome 13 and is widely implicated in cancer [60,61,62]. Overexpression can result from amplification of the miR-17~92 locus [60] or direct miR-17~92 transactivation by c-MYC/*MYCN* [63,64,65]. The oncogenic nature of miR-17~92 activation is supported by the identification of miR-17~92 targets with key roles in cell-cycle control and cell death. In particular, miR-17 and miR-20a target the cyclin-dependent kinase inhibitor CDKN1A (p21), a negative regulator of the G1-S transition, and miR-17 targets the proapoptotic BCL2L11 gene (Bim) [63]. The Ventura team performed a groundbreaking in vivo study to further unravel the role of the individual members of this cluster and found, amongst others, a role for miR-19 in Myc-driven tumorigenesis in two models of human cancer [66]. Investigation of a large series of NB cases revealed one tumor with a small focal gain of the chromosome 13q31.3 region encompassing the miR-17~92 locus as only locus present in this focal gain, further indicating a supportive role for enhanced expression levels of one or more miRs from this cluster and selective pressure during the tumor formation process for those cells carrying an extra copy of the locus [44]. Using combined SILAC and quantitative mass spectrometry, the effects of activation of the miR-17~92 cluster on global protein expression in NB cells were investigated showing cooperation between individual miR-17~92 miRNAs and implicating miR-17~92 in multiple hallmarks of cancer, including proliferation and cell adhesion [65]. In this study, miR-17~92 was found to act as a potent inhibitor of TGF-β signaling through affecting both upstream and downstream regulators of pSMAD2 thus triggering downregulation of multiple key effectors along the TGF-β signaling cascade as well as direct inhibition of TGF-β-responsive genes. Of further note, using an integrated approach including miRNA and messenger RNA (mRNA) gene expression data it was shown that c-MYC/*MYCN* induced, in addition to the miR-17~92 cluster, a broader core set of miRNAs for cooperative repression of common transcriptional programs related to disease aggressiveness [67].

### 3.4. ALYREF

The Marshall team used a bioinformatic approach to identify critical genes on 17q in high-risk NBs by evaluating frequency of gains distal to 17q21.31 using whole genome sequencing data from NB tumors from the US TARGET database [68]. A total of 1044 transcripts from the 17q21-ter locus were analyzed for differential gene expression in 17q21-ter-gain patients (1) and *MYCN*-amplified patients (2) in the same cohort. Further correlation with gene copy number (3) and *MYCN* expression (4) as well as association with poor NB patient outcome (5) were taken into account. Using this approach, *ALYREF* was the only gene to pass all five criteria among the 1044 17q21-ter genes, and suggests that *MYCN* and *ALYREF* co-operate as tumorigenic factors in NB. Further functional studies revealed that *MYCN* increases NB cell viability in an *ALYREF*-dependent manner and directly regulates *ALYREF* transcription. The known role of *ALYREF* as a regulator of DNA binding guided further analyses that uncovered *ALYREF*-*MYCN* interaction in a nuclear coactivator complex which stimulates transcription of the ‘ubiquitin specific peptidase 3′ (USP3), consequently reducing *MYCN* ubiquitination and degradation. This finding therefore opens novel opportunities for targeting this deubiquitinase activity through the development of USP3 inhibitors [68].

### 3.5. MYC

Given that MYC also drives expression of the miR-17~92 locus it would not be unexpected that MYC overexpression could also cause NB and that MYC could be a target for amplification in a subset of NB. We first reported a bona fide NB cell line with evidence for a chromothripsis event involving amongst others multiple chromosome 8q loci including MYC [69]. More recently, the Look team reported amplification of critical enhancers controlling MYC expression in a subset of NBs as well as structural rearrangements leading to enhancer hijacking [70].

**Figure 3 jpm-11-01286-f003:**
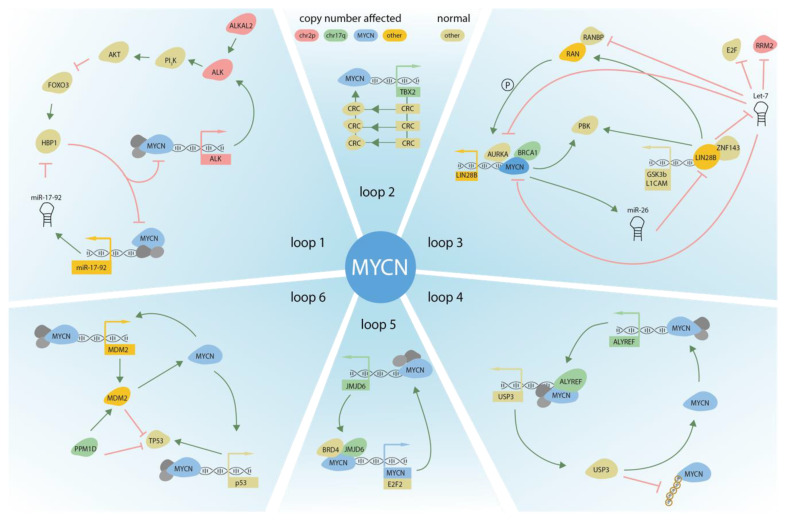
DNA copy number affected genes and their upstream and downstream regulators, supporting *MYCN* activity in NB. Six different autoregulatory loops are depicted. Color of the proteins (circles) or genes (square) indicates the genomic location of the CNA.

## 4. DNA Copy Number Affected Genes Driving Cell Cycle Activity

### 4.1. CDK4/CCND1

Already in 1995, we found evidence for amplification of a region encompassing several putative oncogenes on the long arm of chromosome 12 including *SAS*, *MDM2*, *CDK4*, *GLI*, *CHOP* and *CDK2* [71]. Decrease in *Cdk2* abundance and loss of *CDK4* activity was observed in a mouse NB cell line upon neuronal differentiation in response to DMSO [72]. Gains and amplification of the *CDK4* regulatory subunit ‘cyclin D1′ (*CCND1*) were detected in early DNA copy number profiling efforts of high-risk NBs. In most cases the *CCND1* gains were accompanying large 11q deletions in *MYCN* non-amplified tumors [30]. In a recent whole genome profiling study of 205 high-risk cases Brady et al. confirmed the association of 11q13.3 copy number gains with the chromosome 11 deletions (resulting from unbalanced translocations) resulting in significant increased *CCND1* expression [23].

Dependency on *CCND1* and *CDK4* for NB survival and proliferation was demonstrated and a functional dependency on overexpression of G1-regulating genes to maintain the undifferentiated phenotype was shown [81]. In line with these findings, another study investigated genome-wide copy number alterations and transcriptomes in 82 NB tumors showing that nearly 30% of all tumors had genomic amplifications, gains, or losses with shortest regions of overlap that suggested implication of a series of G1 cell cycle regulating genes. This included the above mentioned *CCND1* and *CDK4* which were amplified or gained and the chromosomal regions containing the *CDKN2* (INK4) group of CDKIs which were frequently deleted. Cluster analysis showed that tumors with genomic aberrations in G1 regulating genes over-expressed E2F target genes, which regulate S and G2/M phase progression [137]. This was further explored in the context of doxorubicin induced DNA damage response. The altered *CDK4*/cyclin D-pRB axis in *MYCN*-amplified NB cells allowed to evade a G1/S arrest after doxorubicin-induced DNA damage. Additional chromosomal aberrations affecting the p53-p21 and *CDK4*-pRB axes further enhance the effects of *MYCN* on the G1 checkpoint and reduce sensitivity to cell death after doxorubicin treatment. *CDK4* inhibition partly restores G1/S arrest and sensitizes cells to doxorubicin-mediated cell death in *MYCN*-amplified cells with an intact p53 pathway [73]. Taken together, these studies identified proteins controlling the critical G1/S restriction point as putative therapeutic targets for high-risk NB [74]. Using an siRNA library for 131 cell cycle regulators, we found *CCND1* and PLK1 as top hits and further investigated the effects of a small molecule compound palbociclib which is a potent and highly selective inhibitor of *CDK4* and *CDK6*. NB cell lines IMR-32, SH-SY5Y, and NGP responded in a time- and dose-dependent manner with reduced proliferation while SK-N-SH and CLB-GA cells were relatively resistant to the treatment and other cell lines such as SH-EP responded at relatively high concentrations [75]. The Maris team evaluated the effect of dual *CDK4*/*CDK6* inhibition on NB viability using the highly specific *CDK4*/*6* inhibitor ribociclib (LEE011, Novartis Oncology, Basel, Switserland) showing the expected reduction in proliferation in 12 of 17 human NB-derived cell lines through cell-cycle arrest and cellular senescence [76]. More recently, a phase I clinical trial was conducted with ribociclib in pediatric cancer patients to test safety and pharmacokinetics with positive results [77]. Next, the combination of ribociclib and the *ALK* inhibitor ceritinib were tested showing a higher cytotoxicity and synergy scores in cell lines with *ALK* mutations as compared with cell lines lacking mutations or alterations in *ALK*. Combination therapy achieved complete sustained regressions in NB xenografts with *ALK*-F1174L and F1245C de novo mutations and prevented the emergence of resistance. Murine ribociclib and ceritinib plasma concentrations were unaltered by the use of combination therapy [78]. Synergistic interaction with ribociclib was also found for MEK1/2 inhibition [79]. In contrast, combined drugging of *CDK4*/*6* and *MDM2* activity did not show noticeable synergistic effects [80].

### 4.2. TRIM37

*TRIM37*, the centrosomal ubiquitin ligase on chromosome 17q, was identified as a top hit in a genome-wide screen for genes whose inactivation enables sustained proliferation of PLK4 inhibitor (centrinone) treated retinal pigment epithelium (RPE1) cells. Loss of *TRIM37* rescued the delayed spindle assembly and chromosome-segregation failure seen in cells that lack centrosomes treated with centrinone. Further analyses revealed that *TRIM37* is an essential determinant of mitotic vulnerability to PLK4 inhibition: low *TRIM37* levels accelerate acentrosomal spindle assembly and improve proliferation following PLK4 inhibition, whereas high *TRIM37* levels inhibit acentrosomal spindle assembly, leading to mitotic failure and cessation of proliferation. NB and breast cancer cells with gain of chr17q were found to be highly sensitive to PLK4 inhibition [82].

## 5. DNA Copy Number Affected Genes Involved in *TP53* Pathway Control

### 5.1. MDM2

The first evidence for *MDM2* amplification in NB was reported by Waber et al. [138] in 1993 and later confirmed by several other studies [71,83,84,85]. As indicated above, multiple other loci are often co-amplified and more complex amplification patterns affecting distinct chromosome 12q chromosomal segments hint towards chromothripsis causing events. The p53 function is inhibited by *MDM2*, and while *TP53* mutations are rather rare in NB, its occurrence is associated with relapse cases [86]. Mutations in the RAS or p53 pathway increases tumor aggressiveness in the high-risk cases defined by telomere maintenance, while this is not the case in low-risk tumors [139]. Multiple studies pointed towards disruption of the *TP53*/*MDM2*/p14 signaling axis in wild-type p53 NB and the functional consequences have been extensively investigated [87,88,89,90]. In addition to its canonical function as p53 inhibitor, the *MDM2* ubiquitin ligase also controls *MYCN* protein stability which itself drives *MDM2* expression through promotor binding and transcriptional activation [91,92,93,94,95]. Importantly, *MDM2* overexpression and dependency offers a target for precision drugging given the potential to disrupt *MDM2*-p53 protein-protein interaction in *TP53* wild type NB cells. Using a low-molecular-weight compound that competes with p53 for binding into a hydrophobic cleft on the surface of *MDM2*, termed nutlin-3 [96], we have been able to elicit a robust and selective activation of the p53 pathway in NB cells with wild-type p53, leading to G1 cell cycle arrest, apoptosis, premature senescence and neuronal differentiation [97]. *MDM2* inhibition by nutlin-3 also dramatically sensitizes NB cells to chemotherapy-induced apoptotic cell death [98], and nutlin-3 can reverse P-glycoprotein-mediated multidrug resistance of NB cells regardless of *TP53* mutation status [99]. These findings prompted further exploration of nutlin-3 derivatives for novel clinical trials in pediatric tumors including NB [100,101].

### 5.2. BIRC5

Survivin, encoded by the *BIRC5* locus is an essential protein for cell division and can inhibit cell death in most, if not all cancers including high-risk NBs. Of further interest, the *BIRC5* gene maps to the commonly gained 17q segment and thus could represent a bona fide target affect by extra 17q copies [102,103]. Given that survivin is the fourth most upregulated mRNA in the human cancer transcriptome [140] and its expression has been correlated with increased tumour resistance to a broad range of chemotherapy agents, radiation insensitivity and poor patient prognosis, it has been investigated as target for therapeutic intervention. Currently, the best-studied survivin suppressor is YM155 (sepantronium bromide) which has been tested with variable results [104]. A recent paper has validated the role of survivin in NB development and describes the potential of utilizing survivin as drug target. It was shown that co-overexpression of *BIRC5* in Tg(dβh:eGFP-*MYCN*) transgenic zebrafish promotes the development and growth of *MYCN*-driven NB (Dolman et al., in review) in keeping with data from Hipp et al. showing that survivin enhanced the tumorigenic functions of *MYCN* in Rat1 fibroblasts [105]. Recently, Michaelis et al. found no difference in efficacy of response to sepantronium bromide between *MYCN* non-amplified and *MYCN*-amplified in vitro NB model systems [106]. In vivo analyses in mice showed strong response to sepantronium bromide for KCNR NB cells [107]. Further efforts to identify potent on-target surviving inhibitors are ongoing including the search for small molecule inhibitors of the survivin dimerisation interface [141], or adjacent cavities [142], antibodies binding to a cell surface pool of survivin [143], and short interfering RNA directing survivin which is encapsulated in nanoparticles [144].

## 6. DNA Copy Number Affected Genes Driving Transcriptional Addiction

### 6.1. TBX2

Given the established important role of super-enhancer marked transcription factor encoding loci in several cancer entities we set out to identify such candidates residing on chromosome 17q. The ‘T-box 2 transcription factor’ (*TBX2*), a gene with unknown function in NB, was prioritized as transcription factor with top-ranked super-enhancer score in NB cell lines and with expression levels highly correlated with survival outcome in NB tumors [109]. Based on integrated analysis of the *TBX2* genome-wide DNA binding pattern and transcriptome analysis upon knockdown, we identified *TBX2* as novel member of the core regulatory circuitry that marks high-risk NBs. Similar findings were obtained using an unbiased genome-scale CRISPR screen identifying 147 candidate essential genes in *MYCN*-amplified NB cell lines, pinpointing *TBX2* as member of the core regulatory circuitry in *MYCN*-amplified NB [108]. Our data suggest a role for *TBX2* in positive control of E2F-FOXM1 signaling and proliferation. We propose that *TBX2*, as member of the core regulatory circuit, further enhances *MYCN* driven proliferation. Combined pharmacological targeting of transcriptional addiction using a BET (JQ1) and *CDK7* (THZ1) inhibitor, yielded synergistic effects on *TBX2* downregulation leading to massive apoptosis.

### 6.2. JMJD6

To further identify transcriptional regulators affected by 17q gains, the Tao team looked for candidates and identified the ‘Jumonji domain-containing 6′ (*JMJD6*) gene which encodes a dual arginine demethylase and lysyl hydroxylase of histone and non-histone proteins. As a histone arginine demethylase, *JMJD6* modulates RNA polymerase II release from promoter-proximal pause regions at target gene anti-pause enhancers by forming a protein complex with BRD4 and demethylating histone H4 at arginine 3 (H4R3) which consequently results in transcriptional activation. Whereas as a lysyl hydroxylase, *JMJD6* also forms a protein complex with p53 and catalyzes p53 protein hydroxylation, which results in p53 inactivation. *JMJD6* was shown to form protein complexes with *MYCN* and BRD4, controlling E2F2 and *MYCN* transcription, supporting proliferation and survival of NB cells. Combination therapy with the *CDK7* (super-enhancer) inhibitor THZ1 and the histone deacetylase inhibitor panobinostat synergistically reduces *JMJD6*, E2F2 and *MYCN* expression, inducing apoptosis in vitro and causing NB tumor regression in mice [110].

### 6.3. SOX11

Our team identified *SOX11* as putative dependency gene through a search for focal gains and/or amplifications of chromosomal segments encompassing transcription factors with a putative or known role in normal (neuronal) development. Re-analysis of NB DNA copy number profiling data revealed the sympatho-adrenal lineage specific *SOX11* gene in recurrent chromosome 2p focal gains and amplifications [111]. *SOX11* is specifically expressed in adrenergic NBs and absent in mesenchymal NBs [111,112], two distinct super-enhancer associated subtypes in NB [135,136]. Indeed, *SOX11* is regulated by multiple adrenergic specific *cis*-acting (super-) enhancers, along with adrenergic NBs being strongly dependent on high *SOX11* expression levels for growth and proliferation [111]. *SOX11* was also shown to be involved in nucleokinesis in adrenergic NB cells [112]. We identified and validated functional *SOX11* target genes through genome-wide DNA-binding and transcriptome analysis and identified several genes which are implicated in chromatin remodeling and epigenetic modification. Given the broad control of *SOX11* on multiple epigenetic regulatory complexes and its presumed pioneer factor function [145], we hypothesize that adrenergic NB cells have co-opted the normal role of *SOX11* as a crucial regulator of chromatin accessibility and cell identity [111].

### 6.4. SOX9

Large cohorts of NB patients have been studied using genome wide association (GWAS) analysis to identify NB predisposing loci. One such locus on 6p22.3 was further investigated and shown to encode *CASC15* and *NBAT1* lncRNAs acting as tumor suppressors with reduced expression in high-risk NBs. These lncRNAs were shown to repress *SOX9* expression levels through CHD7 ([113], p. 9), a known regulator of neuronal differentiation and (super-) enhancer activity [114]. Of further interest, *SOX9* is shown to be regulated by a mesenchymal-specific super-enhancer in NB [136], and plays a role in NB migration and invasion [115]. Given the *SOX9* location on 17q25 which is commonly gained in high-risk NB, it is conceivable that dosage effects of *SOX9* further enhance differentiation arrest during NB formation. Interestingly, CHD7 protein stability is regulated by the *CASC15* and *NBAT1* interactor USP36 which is also located on the chr17q gained region [113].

## 7. DNA Copy Number Affected Genes Controlling DNA Damage and Replicative Stress Response

### 7.1. PPM1D/WIP1

While for other chromosomal regions discussed above, rare but recurrent high-level amplifications were found, for 17q gains recurrently affected amplified regions are extremely rare. Through DNA copy number analysis of a less commonly used NB cell line M-PN-TS high level amplification of a small region encoding 15 genes (seven of which were consistently overexpressed, including *PPM1D* (*WIP1*)) was delineated. Further functional experiments revealed reduced proliferation and survival upon *PPM1D* knockdown [116]. In a recent analysis on 208 tumors with chromosome 17q gain, only three genes revealed to be present in the smallest region of overlap including *PPM1D*, *RAD51C* and *BRIP1*, and these chromosome 17q segmental gains were shown to further accumulate during clonal evolution. Furthermore, a gene fusion of *PPM1D* and *BCAS3* was also found to be present in one NB tumor, subsequently resulting in accumulating *PPM1D* expression levels [117]. *PPM1D* is a negative regulator of p53 and positively regulates *MDM2* thus further enhancing p53 inhibition (Figure 3, loop 6) [117,146]. Using a CRISPR screen, wild-type P53 cell lines were shown to be dependent on *PPM1D* expression and *PPM1D* knockdown delayed tumor formation in vivo. *PPM1D* is also shown to be a critical regulator of DNA damage response by dephosporylating and inactivating ATM, ATR, CHK1, CHK2 and H2AX. A transgenic mouse model with *PPM1D* overexpression exposed to irradiation and thus DNA-damaging stress was able to develop PHOX2B-expressing neural crest-derived primary tumors of the adrenal gland phenotypically and genetically similar to NB [117]. Pharmacological inhibition of *PPM1D* selectively suppressed tumor growth of p53 wild-type NB cell lines [146] and of established NB xenografts in nude mice indicating that *PPM1D* might be a promising therapeutic target in p53 wild-type NBs [117].

### 7.2. BRCA1, BRIP1, CDK12

Three genes located on 17q are involved in the DNA repair pathway and upregulated during TH-*MYCN* driven NB formation in mice [147], possibly playing a role in NB and/or act as potential novel drug targets. *CDK12* was shown to regulate DNA repair genes (including *BRCA1* and *BRIP1*) through intronic polyadenylation [119,120]. Several *CDK12* inhibitors have been developed and are currently under evaluation for clinical trials for other cancers, (p. 12 in [121,122,123]). Of further note, the Winter lab discovered a cyclin K molecular glue degrader which also efficiently depletes *CDK12* and which could also represent a potentially important novel molecule for precision therapies [124].

Interestingly, the Eilers team attributed a novel function to the repertoire of the *BRCA1* gene. *MYCN* was found to recruit *BRCA1* to promoter-proximal regions preventing *MYCN*-dependent accumulation of stalled RNAPII and enhancing transcriptional activation by *MYCN*. *BRCA1* enables *MYCN* to suppress R-loop formation in promoter-proximal regions by stabilizing mRNA decapping complexes. The critical signal in *MYCN* that enables recruitment of *BRCA1* is the dephosphorylation of Thr58, which allows binding of the ubiquitin-specific protease USP11 and stabilizes *MYCN* and *BRCA1* on chromatin, preventing proteasomal turnover of *MYCN* (Figure 3, loop 3) [118]. Mitoxantrone inhibits USP11 thus offering opportunities for targeting the *BRCA1*-*MYCN* interaction [148]. Because *BRCA1* is highly expressed in neuronal progenitor cells during early development and MYC is less efficient than *MYCN* in recruiting *BRCA1*, the Eilers team suggests that a cell-lineage-specific stress response enables *MYCN*-driven tumours to cope with deregulated RNAPII function. *BRIP1* (*BRCA1*-interacting protein or FANCJ) maps in close proximity distal to *PPM1D* and *TBX2* on 17q23.2 and has been implicated in unwinding of stable G-quadruplex DNA structures and acting through multiple mechanisms to the replication fork [116]. Our team is currently investigating the possible role of *BRIP1* in further details including its effects in *MYCN* driven NB formation in a *MYCN*- driven zebrafish model.

### 7.3. RRM2

We recently identified the *RRM2* gene as target for focal gains and amplifications in high-risk NBs, amongst others resulting from chromothripsis [125]. *RRM2* encodes for the ‘ribonucleotide reductase regulatory subunit 2′ (*RRM2*) and together with RRM1 forms the RNR ribonucleotide reductase enzyme that ensures dNTP production essential for DNA replication and repair. *RRM2* has been proposed as driver oncogene in prostate cancer [126] and has been shown to play a critical role in melanoma [127] and Ewing sarcoma [128]. Based on its implication in focal amplifications and gains together with strong upregulation in development of TH-*MYCN* driven NB in mice and inverse correlation with survival in NB patients, our team further investigated the role of *RRM2* through in vitro and in vivo experiments [125]. NB cell lines were shown to strongly depend on *RRM2* for survival. Furthermore, a zebrafish *MYCN*/*RRM2* co-overexpression model revealed a strong increase in tumor penetrance compared to *MYCN* only overexpressing animals. Finally, we demonstrated synergism for combined *RRM2* and CHK1 inhibition thus pointing towards this drug vulnerability as an important future angle for therapeutic exploration [125].

## 8. Evaluation of Gene Dependencies and Candidate Therapeutic Targets Using the Cancer Dependency Map

In order to prioritize for essential genes and candidate therapeutic targets present on copy number affected lesions, NB tumor dependency has been investigated both in vitro and in vivo, as discussed above. However, public datasets can offer a first insight to prioritize candidate genes based on their association with survival and risk groups, other copy number affected genes, or tumor dependency. Using a recent public available CRISPR screen across 1032 cancer cell lines including 31 NB cell lines, genes can be evaluated for their NB dependency and specificity [149,150,151]. Figure 4 depicts the chromosomal locations and dependency scores based on this CRISPR screen for all the copy-number affected genes discussed in this review. For all of them, dependency is shown in NB. Of further interest, *ALK*, *LIN28B*, *CCND1*, *PPM1D*, *TBX2*, *SOX11*, *MYCN* and *BRIP1* are strongly selective for NB as compared to other tumor entities, further highlighting these genes as potential drug targets (Figure 4). Recently, Dharia et al. created a first-generation Pediatric Cancer Dependency Map for 13 pediatric solid and brain tumor types, for which a full genomic characterization of 26 NB cell lines and 180 NB tumors was performed amongst other cellular entities and matched with genome-scale CRISPR-Cas9 loss-of-function screens. They showed that the number of selective dependencies does not correlate with the number of mutations or copy number alterations. These data could further be used to identify potential biomarkers for individual genetic dependencies, e.g., selective *MDM2* dependency in *TP53* wild-type cells. The vulnerabilities seen in childhood cancer were often different than what is observed in adult cancer, indicating that adult oncology drugs will be most of the times insufficient to treat pediatric cancers [152].

## 9. Current and Emerging In Vitro and In Vivo Models for Exploring the Role of DNA Copy Number Driven NB Dependencies

Both mouse and zebrafish models have been established for *MYCN*-driven NB formation [22,46,47,48,50,54,59]. In particular the zebrafish model has proven to be an efficient tool for testing co-dependencies as illustrated by successful studies for multiple candidates including mutant *ALK* [48], LMO1 [153], *LIN28B* [59], *RRM2* ([125], p. 2), *BIRC5* (Dolman et al., in review) and *BRIP1* (Vanhauwaert et al., in preparation). One particular advantage of zebrafish models is access to early developing lesions that can be isolated by FACs sorting [154] and which is currently under further investigation in our lab for deeper exploration of early transforming events using single cell technologies. In addition to these approaches, the Freeman team has followed a unique approach to isolate immature neural crest cells from mouse embryos [155]. Subsequently, following transduction of these cells with a *MYCN* overexpression construct and injection into nude mice, it was shown that in most instances NB develop that recapitulate much of the molecular and cellular phenotype of mouse and human NBs, including CNAs syntenic as those observed in human e.g., 17q gain, 2p gain and 1p36 deletion. This model was also exploited to further study the role of ARID1A in the process of early tumor formation [156], a locus previously shown to be affected by deletions and alterations in NB [157], by CRISPR mediated induction of deletions encompassing the ARID1A locus [156]. In contrast to this model where the cells are subcutaneously injected into the flank of mice, Cohen et al. developed a NB model system by micro-injecting human pluripotent stem cell-derived NC cells with conditional *MYCN* and *ALK* expression in utero into developing mouse embryos. These mouse-human neural crest chimeras developed tumors resembling human NB upon activation of the oncogenes by adding doxycycline to the drinking water [158]. So far, no one was able to engineer cells or animals with large chromosomal segmental gains for in vivo evaluation. However, using an in vitro embryonal stem cell model we could show that recurrent chromosomal segmental gains, including chromosome 17q amongst others, provide a proliferative advantage when cells are under increased replicative stress [159]. In vitro differentiation outside the developmental context of the normal neural crest is challenging, particularly for cells migrating during differentiation. Thus far, current in vitro models have been unable to reliably generate sympatho-adrenergic precursors, from which NB is presumed to develop. Our lab, in collaboration with the Studer laboratory at Memorial Sloan Kettering Cancer Center, has optimized and characterised a pluripotent stem cell-derived in vitro differentiation model to accurately generate the cells of interest. This allows us to study the normal sympathetic as well as NB development in vitro, using any available stem cell line carrying mutations of interest. We are currently utilizing this model to study *MYCN*-driven NB formation, where we induce *MYCN* overexpression during in vitro development to analyse the effect on development and malignant cell transformation. Using the 17q gained-stem cell line available in our lab, we are also in the process of studying the role of the 17q gain in *MYCN* driven NB. These state-of-the art experimental platforms are highly valuable to study NB tumor initiation, progression, manifestation, tumor-immune-system interaction and drug response in depth.

## 10. Future Perspectives to Mine CNAs for Novel Candidates and Deeper Exploration of NB Dependency Genes

Computational approaches have been developed and applied to mine genome-wide CNAs and gene expression profiles in order to identify novel tumor dependencies. One of the first algorithms, ‘Copy Number and Expression In Cancer’ (CONEXIC), is a module networks-based approach that integrates matched CNA and gene expression data from tumor samples to identify and score the combination of CNAs that best explain the behavior of a gene co-expression module across tumor samples [160]. CONEXIC correctly identified known drivers of melanoma and predicted multiple novel ones, some of which could be empirically confirmed. Meanwhile, several related methods have been reported that can integrate various multi-omics data and outperform CONEXIC both in computational runtime and ease of parameter setting, as well as in the prediction of consistent and functionally coherent co-expression modules and associated driver genes i.e., the regulators of the co-expression modules [161,162]. In this respect, Lemon-Tree is an integrative multi-omics network inference algorithm where the learning of the co-expression modules and the regulator assignment are decoupled and repeated multiple times in order to construct a consensus solution for both. Based on a large dataset of somatic CNA and gene expression profiles from The Cancer Genome Atlas, Lemon-Tree correctly identifies known glioblastoma oncogenes and tumor suppressors as master regulators in the inferred module network [161]. However, matching CNA and gene expression profiles are not always available, and CNAs do not always translate proportionally into altered expression levels due to transcriptional adaptive mechanisms and due to the fact that they exert effects further downstream on core tumor pathways through densely connected gene regulatory networks [163,164]. Therefore, several studies have shown that the effect of genomic alterations in cancer and the search for novel tumor dependencies can be more readily assessed by the inference of gene regulatory networks based on aggregated datasets of gene expression profiles that is specific for the cancer type under study [165,166,167]. Other information, in the form of regulator binding (e.g., ChIP-sequencing data for transcription factors) or multi-omics can be integrated in the process, for a more causal inference of regulatory interactions and a more significant identification of driver genes [168]. ModuleOmics is a statistical framework that simultaneously integrates correlated expression, transcriptional coregulation, protein-protein interactions and mutual exclusivity of mutations and CNAs into highly functionally connected modules enriched with cancer driver genes, outperforming state-of-the-art single omics approaches [169]. Another recently developed tool, Moonlight, also incorporates multi-omics data for the discovery of tumor suppressors, oncogenes and dual role genes in the framework of gene regulatory networks using manually curated expert knowledge and/or machine learning on contrasting normal and tumor samples. While gene expression data are used as the major source of information to detect candidate driver genes, a second layer of evidence from another genome or epigenome data modality e.g., hypermethylation is required to define the candidate drivers as critical cancer driver genes [170].

The transcriptional network-based methods described above have already been successfully applied to NB. Consensus clustering of the TARGET NB patient cohort revealed three molecular subtypes of high-risk NB, consistent with genomic aberrations: *MYCN*-amplified status and 1p36 deletion for cluster 1, 11q deletion for cluster 2 and a mesenchymal signature without any strong genomic aberration association or cluster 3. Subsequently, master regulators for each of the high-risk subtypes were inferred using the VIPER and ARACNe-AP algorithms based on the enrichment of their transcriptional target genes in the subtype specific gene expression signatures. As an example, a TEAD4-*MYCN* positive feedback loop emerged as a regulatory driver of the *MYCN*-amplified high-risk NB subtype [171]. A similar approach was used on gene expression profiling data obtained during murine TH-*MYCN*-driven NB tumor formation at different time points revealing *MEIS2* as candidate NB tumor-initiating factor [147]. In another study, weighted gene co-expression network analysis on publicly available microarray data of 206 NB patients and subsequently association of these modules to chromosomal alterations, indicated that modules involved in nervous system development and cell cycle are highly associated with *MYCN* amplification and 1p deletion, while modules responding to immune system process are associated with *MYCN* amplification only. *BUB1B* and *CD53*, which serve as hub genes in modules responding to cell cycle and immune system processes respectively, are put forward as potential novel drivers. Also from this analysis, *CADM1* emerges as a top tumor suppressor gene candidate in 11q-deleted NB [172]. A deep learning auto-encoder framework integrated CNAs and gene expression profiles from the TARGET and SEQC NB cohorts and after filtering for survival dependent genes and/or CNAs, this was combined with k-means clustering to detect two subtypes with significant survival differences [173]. In the context of NB, the gene regulatory networks of the developing neural crest are also worth investigating. Using chromatin and transcriptional profiling of cranial neural crest at population and single cell level, followed by network inference and data integration, the gene regulatory networks of the neural crest was charted [174].

Currently, the potential for adding multiple additional omics layers to copy number, gene expression and methylation is rapidly expanding such as chromatin accessibility (using ATAC-seq), epigenetic status (using ChIP or CUT&RUN/TAG for chromatin marks) and proteomics [175]. Moreover, single cell technologies are rapidly evolving and offering further opportunities for deep data mining approaches towards more powerful detection of tumor dependencies. Also, approaches are being developed to predict synthetic lethal interactions which may also open further possibilities for target prioritization [176].

## 11. Conclusions

Insights into NB biology and behavior have come a long way ever since the discovery of *MYCN* amplification. In this review, we have provided a summary of our current insights into copy number driven dependencies. In view of the limited search space for precision oncology drug targets due to the low mutation burden of NBs, further approaches to identify novel druggable vulnerabilities are critically important if we want to increase survival rates significantly. Together with current developments of novel approaches to target so-call undruggable targets such as protein degraders [177], we expect a significant number of novel venues to open towards development of less toxic drugs with higher efficacy to eradicate tumor cells. Finally, these developments can be expected to synergize and integrate into emerging novel insights how NB escapes the host immune system and strategies to transform the immunologically cold high-risk NBs into hot tumors that will be more amenable to emerging and new immunotherapeutic treatment regimes.

## Figures and Tables

**Figure 1 jpm-11-01286-f001:**
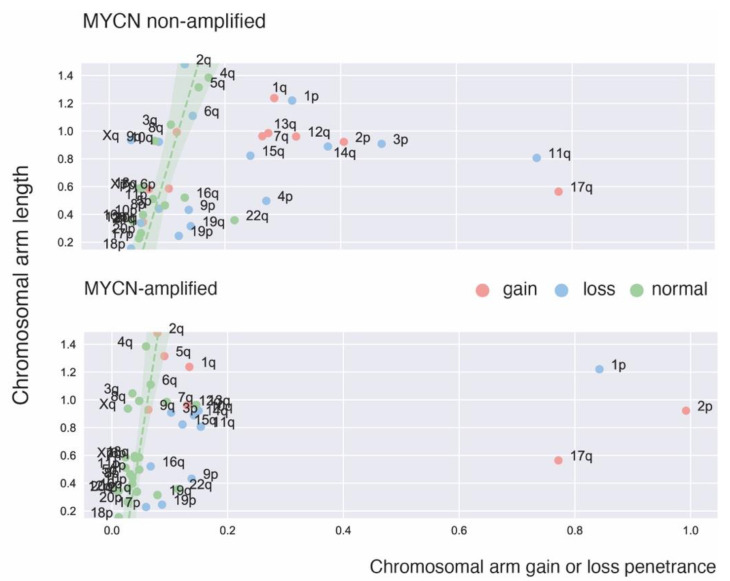
Correlation between chromosome arm length and amount of gains and/or losses on the arm in a dataset of 542 high-risk primary NB tumors [38] showing recurrent copy number imbalances in *MYCN* non-amplified and *MYCN*-amplified tumors. Gains or losses that spanned both chromosomal arms where not withhold. For every chromosomal arm the ratio of gains versus losses was analyzed in a binomial test to establish whether either the number of gains was exceptional in respect to the number of losses or the other way around. If a significant amount of gains was found for the total number of gains and losses, the chromosomal arm is displayed in red, if the other way around in blue, and if both losses and gains on the arm were on par (*p*-value binomial test ≥0.05) they are displayed in green. The ‘green’ arms were used to calculate a linear regression between chromosomal arm length and total number of combined losses and gains. The standard deviation on this linear regression is depicted in a shaded green area to indicate where normal amounts of gains and losses lie for the chromosome arms of different length. Outliers can now easily be identified in the two *MYCN* groups and comprise the usual suspects.

**Figure 4 jpm-11-01286-f004:**
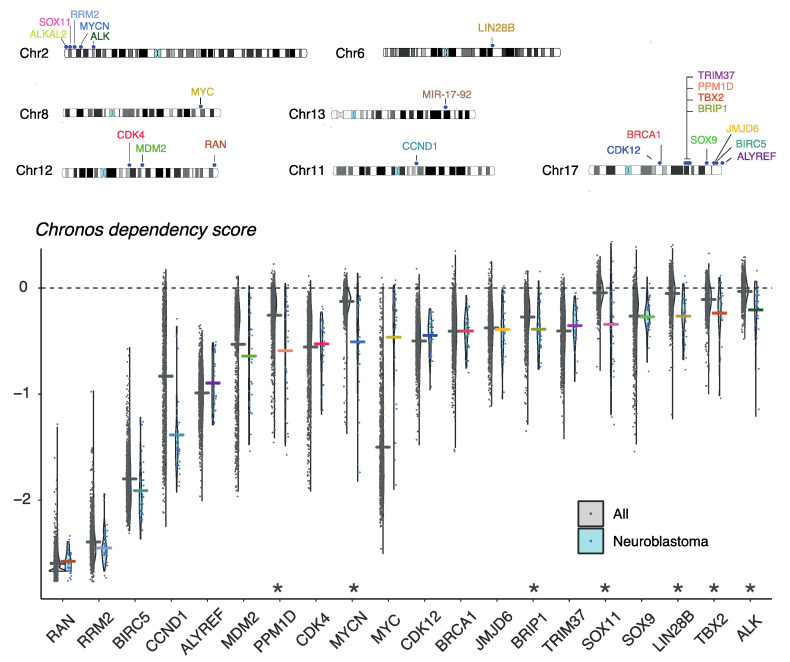
Copy number affected genes in NB that were discussed in this review are depicted on their chromosome (chromosome 2, 6, 8, 11, 12, 13 and 17). Chronos dependency score for each copy number affected gene discussed in this review in a publicly available CRISPR screen across 1032 cancer cell lines (AVANA CRISPR 21Q3, available via the DepMap Portal). A lower Chronos score indicates a higher likelihood that the gene is essential for the cell line. Asterix means that the mean dependency score of the gene in NB cells (blue) is significantly lower than the mean dependency score in other cell lines (grey). Two-group comparison with *p*-value for each gene computed from empirical Bayes moderated t-statistic, enriched lineages are those with *p*-value < 0.0005. Dependency scores for *ALKAL2* and MIR-17-92 are not available.

## Data Availability

The data presented in this study are openly available in GSE103123 and FigShare at https://figshare.com/collections/Meta-mining_of_copy_number_profiles_of_high-risk_neuroblastoma_tumors/4169864 (accessed on 20 November 2021) or through the shiny app https://padpuydt.shinyapps.io/check_cn_in_hr_nb/ (accessed on 20 November 2021) [33].

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
