# Peer review of "From DNA Copy Number Gains and Tumor Dependencies to Novel Therapeutic Targets for High-Risk Neuroblastoma"

_jpm, 2021, doi:10.3390/jpm11121286_

Round 1

Reviewer 1 Report

The content of the review is understandable and the diagrams are good and understandable if you read the text.

As an order, it would be easier to understand at a glance if the role of each Gene is summarized in a Table.

For example

Gene/chromosome locus/main function/representative papers.

Author Response

Point 1: As an order, it would be easier to understand at a glance if the role of each Gene is summarized in a Table. For example Gene/chromosome locus/main function/representative papers.

Response 1: We thank the reviewer for the good suggestion and added a table where we describe the copy number affected gene, chromosomal location, cytoband, main function, in vivo over expression model and drug on target, with the accompanying references. We combined this table with the figure showing the chromosomal aberrations for each chromosome containing copy number affected genes described in the review. 

In addition, the link with the pediatric cancer dependency map published by Dharia et al. (https://pubmed.ncbi.nlm.nih.gov/33753930/) was missing. This was included now. Added text:

Recently, Dharia et al. created a first-generation Pediatric Cancer Dependency Map for 13 pediatric solid and brain tumor types, for which a full genomic characterisation of 26 NB cell lines and 180 NB tumors was performed amongst other cellular entities, and matched with genome-scale CRISPR-Cas9 loss-of-function screens. They showed that the number of selective dependencies does not correlate with the number of mutations or copy number alterations. These data could further be used to identify potential biomarkers for individual genetic dependencies,  e.g. selective MDM2 dependency in TP53 wild-type cells. The vulnerabilities seen in childhood cancer were often different than what is observed in adult cancer, indicating that adult oncology drugs will be most of the times insufficient to treat pediatric cancers[154].

Reviewer 2 Report

This review provides a comprehensive analysis on genetic alterations and copy number changes in particular genes related to neuroblastoma biology.

I congratulate authors on so thorough work, which I believe will be very helpful in the field and provides a nice synthesis of all important players that had been altered and proved important in NB biology. I literally do not have any concerns about this work.

Author Response

We are very pleased to have received a favorable review of our manuscript, thank you.